J Physiol 600.21 (2022) pp 4623–4632

REVIEW-SYMPOSIUM

# Optogenetic approaches to therapy for inherited retinal degenerations

Samantha R. De Silva[1,2,3] and Anthony T. Moore[2,4]

[1] Oxford Eye Hospital, Oxford, UK
[2] UCL Institute of Ophthalmology, London, UK
[3] Nuffield Department of Clinical Neurosciences, University of Oxford, Oxford, UK
[4] Ophthalmology Department, University of California, San Francisco, CA, USA

Handling Editors: Laura Bennet & Omar Mahroo

The peer review history is available in the Supporting information section of this article (https://doi.org/10.1113/JP282076#support-information-section).

**Abstract**  Inherited retinal degenerations such as retinitis pigmentosa (RP) affect around one in 4000 people and are the leading cause of blindness in working age adults in several countries. In these typically monogenic conditions, there is progressive degeneration of photoreceptors; however, inner retinal neurons such as bipolar cells and ganglion cells remain largely structurally intact, even in end-stage disease. Therapeutic approaches aiming to stimulate these residual cells, independent of the underlying genetic cause, could potentially restore visual function in

**Samantha de Silva** is a consultant ophthalmologist at the Oxford Eye Hospital and honorary research associate at the University of Oxford and UCL Institute of Ophthalmology. After graduating from the Universities of Cambridge and Oxford, she was awarded a Wellcome Trust fellowship and undertook prize-winning research into optogenetic gene therapy. This was followed by two subspecialist fellowships in inherited retinal diseases at the Oxford Eye Hospital and Moorfields Eye Hospital. **Anthony Moore** is an Emeritus Professor of ophthalmology at UCSF School of Medicine, having previously held the Michel Vilenski Endowed Chair in Ophthalmology at UCSF. Before moving to UCSF in 2014, he held the Duke-Elder Chair the UCL Institute of Ophthalmology, London. He was previously lead of the inherited eye disease service at Moorfields Eye Hospital. His clinical and research interests are in inherited eye disease, particularly those affecting the retina. He was elected to the UK Academy of Medical Sciences in 2005.

This review was presented at the Physiology 2021 symposium 'Photoreceptors in Health and Monogenic Disease: Advances in Understanding Physiology and Treating Pathophysiology' on 16 July 2021, organised by Dr Omar Mahroo, UCL Institute of Ophthalmology, UK

The Journal of Physiology

patients with advanced vision loss, and benefit many more patients than therapies directed at the specific gene implicated in each disorder. One approach investigated for this purpose is that of optogenetics, a method of neuromodulation that utilises light to activate neurons engineered to ectopically express a light-sensitive protein. Using gene therapy via adeno-associated viral vectors, a range of photosensitive proteins have been expressed in remaining retinal cells in advanced retinal degeneration with *in vivo* studies demonstrating restoration of visual function. Developing an effective optogenetic strategy requires consideration of multiple factors, including the light-sensitive protein that is used, the vector and method for gene delivery, and the target cell for expression because these in turn may affect the quality of vision that can be restored. Currently, at least four clinical trials are ongoing to investigate optogenetic therapies in patients, with the ultimate aim of reversing visual loss in end-stage disease.

(Received 7 March 2022; accepted after revision 18 July 2022; first published online 31 July 2022)
**Corresponding author** S. de Silva: Nuffield Department of Clinical Neurosciences, Level 6 West Wing, John Radcliffe Hospital, Headley way, Headington, Oxford OX3 9DU, UK. Email: samantha.desilva@ndcn.ox.ac.uk

**Abstract figure legend**. Optogenetic approaches for vision restoration in end-stage retinal degeneration. In inherited retinal degenerations, there is progressive loss of photoreceptors (rods and cones) but inner retinal structures remain largely intact. Optogenetic strategies aim to induce light sensitivity in remaining retinal cells using gene therapy, via ectopic expression of an opsin gene. The most widely investigated approach uses retinal transduction via an adeno-associated virus injected either into the subretinal space or via intravitreal injection, with the aim of vision restoration (GC, ganglion cell; BC, bipolar cell; AC, amacrine cell; HC, horizontal cell; RC, residual cones; RPE, retinal pigment epithelium).

# Introduction

The inherited retinal degenerations are a heterogeneous group of disorders that result in photoreceptor cell dysfunction and cell death, leading to severe visual impairment. They affect around one in 4000 people and, with rare exceptions, there are no effective therapies (Verbakel et al., 2018). They may be inherited as an autosomal dominant, autosomal recessive or X-linked trait and some rare forms are associated with mutations in mDNA. Although most of these diseases are confined to the eye, some are associated with systemic features, when the causative gene is expressed in several different extraocular tissues. Interestingly some mutations in ubiquitously expressed genes can give rise to a retina only phenotype.

More than 250 different causative genes have been identified (https://sph.uth.edu/retnet/sum-dis.htm) and this great heterogeneity has complicated attempts to develop effective treatment. Many different approaches have been tried, including the use of gene targeted therapies, neuroprotective agents, optogenetics, cell transplantation, stem cell therapy and retinal implants (Dias et al., 2018). Gene targeted therapies rely on a precise molecular genetic diagnosis and generally will be effective only for the gene of interest, with some approaches utilizing CRISPR or antisense oligonucleotides targeting only a specific mutation (Nuzbrokh et al., 2021). For a disorder caused by more than 250 genes and with much allelic heterogeneity, such targeted approaches have

serious limitations. Furthermore, gene-based therapies are probably effective early in the disease when significant numbers of viable photoreceptors are still present and are probably less effective in advanced disease when there is extensive photoreceptor cell death or photoreceptor structure has been lost. However, many patients have severe disease at presentation and, given that there has been little effective therapy to date, there are many patients with late-stage disease.

Significantly more patients could be helped by the development of treatments that are less influenced by the specific genetic cause and which could still work in late-stage disease to restore visual function. Evidence from animal models (Mazzoni et al., 2008) and human post-mortem specimens (Humayun et al., 1999; Santos et al., 1997) indicates relative preservation of inner retinal structures and circuitry even in advanced stages of inherited retinal degeneration. Several gene agnostic approaches have been investigated for the purpose of vision restoration, including cell transplantation, stem cell therapy, retinal implants and optogenetics. Each has reached the stage of clinical trials after extensive preclinical studies. Cell transplantation and stem cell therapy aim to support residual viable photoreceptors and develop new populations of light sensitive cells. By contrast, optogenetic approaches and retinal implants bypass host photoreceptors to stimulate remaining inner retinal neurons and post-receptoral pathways. The latter comprise arrays of electrodes or photodiodes that are

surgically implanted in the subretinal space or on the surface of the retina permitting direct electrical stimulation of second- or third-order retinal neurons (Bloch et al., 2019; Gekeler et al., 2018). This symposium review will focus on attempts to use optogenetics to restore sight in late-stage retinal degenerations.

## Optogenetics

Optogenetics is a method of neuromodulation that has wide applicability in neuroscience. It involves the use of light to activate a population of cells made photosensitive via ectopic expression of an opsin protein (Zemelman et al., 2002). The first reports utilised the expression of a combination of *Drosophila* photoreceptor genes to induce light sensitivity in mammalian neurones (Zemelman et al., 2002) and several other optogenetic tools have subsequently been developed. Given that the conversion of light to electrical activity occurs in the eye when light triggers phototransduction in rods and cones, an intuitive application of optogenetic techniques is to induce light sensitivity in the remaining inner retinal cells in end-stage photoreceptor degenerations to restore vision. Developing an effective optogenetic approach requires consideration of multiple factors, including the characteristics of the light-sensitive protein used, the vector for gene delivery and the target cell for expression.

## Optogenetic tools

A range of photosensitive proteins have been investigated as optogenetic tools including microbial and mammalian opsins, as well as chimeric and engineered proteins. These vary in their sensitivity to light, wavelength of light causing peak stimulation, response kinetics and potential to cause an immune response when ectopically expressed in the human eye. These proteins can be broadly grouped into ion channels or ion pumps and G-protein coupled receptors.

## Ion channels

The use of light-gated ion channels or pumps is advantageous for the purpose of vision restoration because these do not require the presence of proteins involved in a signalling cascade that may be absent, or reduced, in advanced retinal degeneration. Channelrhodopsin-2 (ChR2) is a rhodopsin with a microbial-type chromophore that was originally isolated from the green alga *Chlamydomonas reinhardtii* (Nagel et al., 2003; Sineshchekov et al., 2002). ChR2 is itself a light-sensitive cation channel: incident light isomerises its *all-trans* chromophore into 13-*cis* retinal inducing a conformational change in ChR2, allowing cations into the cell leading to depolarisation (Nagel et al., 2003). This response is rapid with an onset of less than 50 $\mu$s after light exposure (Holland et al., 1996), making ChR2 well-suited for use as an optogenetic tool. ChR2 responds to light of wavelength under 540 nm, with a peak sensitivity of 450 nm (Nagel et al., 2003), which is in the blue light spectrum.

A disadvantage of ChR2 is the high stimulus intensity required to obtain a response compared to rhodopsin or cone opsins ($10^{15}$ photons cm$^{-2}$ s$^{-1}$ required for ChR2 *vs.* $10^{10}$ photons cm$^{-2}$ s$^{-1}$ for cones and $10^6$ photons cm$^{-2}$ s$^{-1}$ for rods) (Bi et al., 2006; Lagali et al., 2008) and this high intensity of blue light may be damaging to the retina. Another potential limitation is range: healthy photoreceptors are able to respond to light intensities spanning 8 log units, whereas ChR2 responses are seen over a 2 log unit range (Lagali et al., 2008). Therefore, an image or light processing device may be required to amplify dim light, or attenuate very high intensity light for optimal stimulation of ChR2.

Given these limitations, engineered variants of ChR have been developed such as calcium translocating channel rhodopsin (or CatCh), which has improved light sensitivity and kinetics compared to ChR (Kleinlogel et al., 2011). Other native (e.g. ChrimsonR) and engineered ChR (e.g. red-light activated depolarizing ChR or ReaChR) have peak sensitivity in the red light spectrum, and high light intensities of these wavelengths may be less damaging to the retina than similar intensities of blue light. Studies have also explored the use of an engineered light-gated ionotropic glutamate receptor (LiGluR). This bears a mutated cysteine residue, to which a photo switchable tethered ligand can bind. Incident light of 380 nm causes isomerisation of the photo switch and opening of the ion channel, and 500 nm light results in closing of the channel. Expression of this optogenetic tool in murine degenerate retina resulted in restoration of visual responses (Caporale et al., 2011) and a second-generation ligand activated at 460 nm has also shown similar efficacy in mice and dogs (Gaub et al., 2014).

## Ion pumps

Halorhodopsin (NpHR) is a light-sensitive chloride ion pump derived from *Natronomonas pharaonis* halobacteria (Zhang et al., 2007). Light stimulation of NpHR causes hyperpolarisation of the cell (similar to the response of photoreceptors in the physiological state) and its peak sensitivity is to a wavelength of 590 nm. Given its very fast activation and de-activation kinetics (Han & Boyden, 2007), halorhodopsin has good temporal characteristics for therapeutic use. An enhanced variant (eNpHR) has also been developed that permits higher expression levels without inducing toxicity

(Gradinaru et al., 2008). However, the stimulus intensity required to obtain a light response is even higher than that for ChR2 ($10^{16}$ photons cm$^{-2}$ s$^{-1}$) (Busskamp et al., 2010). Other ion pumps have also been investigated such as the yellow–green light sensitive proton pump archaerhodopsin or Arch (Chow et al., 2010), activation of which leads to neuronal silencing. A further tool is Jaws, a red shifted cruxhalorhodopsin, which showed greater light sensitivity and ganglion cell spiking compared to other hyperpolarising ion pumps following expression in residual cone photoreceptors in a mouse model of retinal degeneration (Chuong et al., 2014).

### G-protein coupled receptors

Rhodopsin and cone opsins are found natively in rod and cone photoceptors respectively and activation of each leads to cell hyperpolarisation. They are G-protein coupled, and this has the significant advantage of permitting signal amplification. For example, a single activated rhodopsin interacts with more than 800 molecules of its G-protein transducin, resulting in the closure of more than 200 ion channels in the photoreceptor outer segment disc membrane, from the absorption of one photon alone (Purves et al., 2001). This signal amplification means that G-protein coupled opsins are sensitive to lower light intensities than ion channels, and therefore could function as optogenetic tools (Berry et al., 2019; Cehajic-Kapetanovic et al., 2015) without the need for modification of the light stimulus required by ChR2 and NpHR. In addition, ectopic expression of these native proteins has a lower probability of incurring an immune response when injected into the eye than microbial opsins.

A further mammalian opsin, melanopsin, has also been investigated, which is present in a subset of mammalian cells named intrinsically photosensitive ganglion cells. Melanopsin is also G-protein coupled and photoisomerisation of 11-*cis* retinal results in activation downstream signalling cascade causing cell depolarisation, with responses in the physiological light range (De Silva et al., 2017; Lin et al., 2008). It may also be capable of coupling to ubiquitous cell signalling pathways (Hankins et al., 2008) and therefore probably functional when introduced into a new cell type. However, a major disadvantage of melanopsin is its slow kinetics compared to ChR2 and NpHR (Do et al., 2009) because its response duration is about 20 times that of mouse rods.

To leverage the advantage offered by G-protein coupled receptors, engineered proteins have also been developed such as a chimera comprising domains of the metabotropic glutamate receptor mGluR6 (found in ON bipolar cells) and melanopsin (Opto-mGluR6) (van Wyk et al., 2015) and a photo switch activated metabotropic glutamate receptor 2 (mGluR2) (Berry et al., 2017).

### Target retinal cells

An important consideration in developing an optogenetic approach is whether to target expression of the photo-sensitive protein to a specific retinal cell type or adopt a ubiquitous approach, where the light-sensitive protein is expressed in as wide a range of cells as possible (Fig. 1).

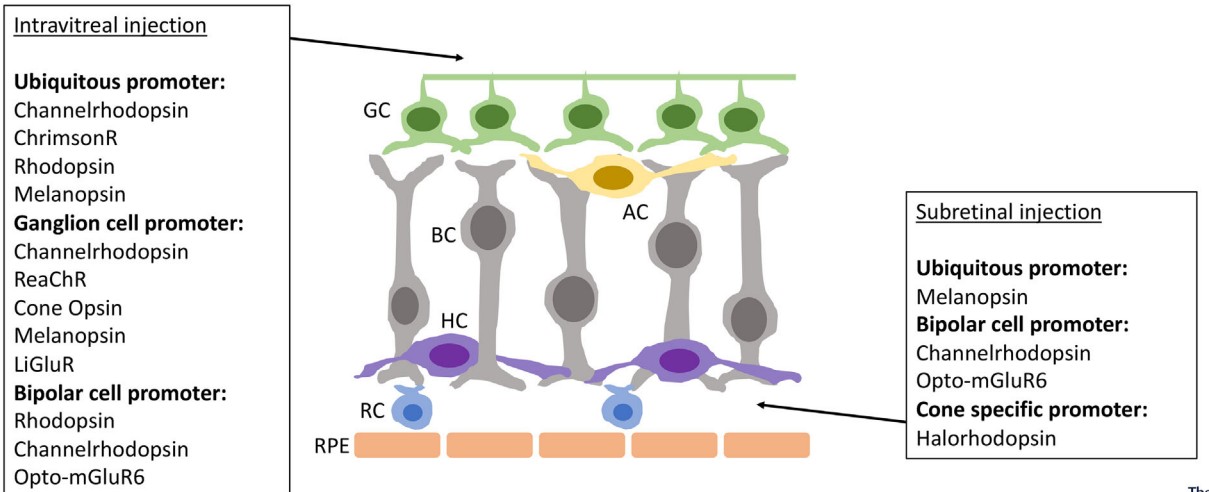

**Figure 1. Optogenetic approaches in advanced retinal degeneration**
Schematic illustrating residual cells in advanced retinal degeneration, showing opsin proteins explored as optogenetic therapies and retinal cells targeted, via gene therapy delivered by intravitreal injection and subretinal injection. GC, ganglion cell; BC, bipolar cell; AC, amacrine cell; HC, horizontal cell; RC, residual cones; RPE, retinal pigment epithelium.

The latter has been widely investigated subsequent to the first study demonstrating efficacy using ubiquitous expression of ChR2 in a mouse model of retina degeneration (Bi et al., 2006); multiple other studies have also demonstrated similar efficacy (Dalkara et al., 2013; De Silva et al., 2017; Lin et al., 2008; Tomita et al., 2007). The rationale here is that the mixed visual signal is interpreted via cortical processing, as is presumed to be the case with retinal implants that have also shown efficacy despite stimulating multiple retinal cell types (Mills et al., 2017).

However, widespread expression of a light sensitive protein in remaining retinal cells runs the potential risk of a loss of signal processing, limiting the quality of vision that could be restored. Human retinal circuitry is complex: cones synapse with 10 types of bipolar cell (ON and OFF subtypes) and rods with one (ON subtype, the most common bipolar cell type in the retina). This multiplicity of connections allows for image processing and the creation of ON and OFF pathways, where a light stimulus excites one population of cells and inhibits another. These connections are further modified by horizontal cells to mediate contrast sensitivity. Cone ON bipolar cells synapse with ON ganglion cells, and OFF bipolar cells synapse with OFF ganglion cells, respectively. Rod ON bipolar cells synapse with amacrine cells, which in turn provide inhibitory input to OFF ganglion cells. There are at least 25 types of amacrine cell and more than 17 types of ganglion cell in the primate retina, with varying functions that facilitate our complex visual responses (Grunert & Martin, 2020).

Some optogenetic strategies therefore aim to preserve as much signal processing as possible within the retina, by targeting the most upstream remaining cells (e.g. bipolar cells or residual cones) to more closely mimic the retinal circuitry that is normally stimulated by light. *In vivo* studies using murine models of retinal degeneration have demonstrated successful targeting of eNpHR to hyperpololarise residual cones (Busskamp et al., 2010) and ChR2 to depolarise ON bipolar cells (Doroudchi et al., 2011; Lagali et al., 2008). However, hyperpolarising opsins (e.g. rhodopsin) have been also been selectively expressed in ON bipolar cells in mouse models with restoration of visual responses (Cehajic-Kapetanovic et al., 2015). Precisely how the visual signal would be interpreted in the latter situation is unclear, given reversal of polarity, although there is evidence to suggest that visual responses were mediated via light-dependent disinhibition of ganglion cell firing (Cehajic-Kapetanovic et al., 2015). Alternative strategies specifically target the ganglion cell (Berry et al., 2019; Sengupta et al., 2016) because this cell type is preserved even in end-stage disease, but again compromise on bypassing the signal processing capacity within the retina.

## Delivery of optogenetic agents to the retina

The most widely investigated approach for delivery of the opsin protein to cells of the degenerate retina is via gene therapy vectors such as adeno-associated viral vectors (AAV). These vectors have been used in human gene supplementation therapy clinical trials and in a Food and Drug Administration and European Medicines Agency approved retinal gene replacement therapy named voretigene neparvovec (Luxturna), for an early-onset, severe retinal degeneration (as a result of biallelic variants in *RPE65*) (Maguire et al., 2019). AAV are single-stranded DNA parvoviruses that are not associated with pathogenicity in humans, and have a genome of ~4.7 kb consisting of two genes on either side of which lies a palindromic region known as an inverted terminal repeat (ITR). These native genes can be replaced with a transgene of interest and regulatory elements, such as the coding sequence of an opsin gene. A limitation of AAV gene therapy is the size of genetic material than can be packaged within the AAV capsid, which is under 5 kb. However, given that opsin genes are comparatively small, this limited packaging capacity does not significantly impact optogenetic gene therapy approaches.

The AAV vector can be further altered to influence efficacy of expression and target cell type by modifying its capsid and promoter sequence amongst other elements. More than 100 AAV serotypes have been described (Lebherz et al., 2008), of which AAV2, AAV3 and AAV5 are endemic to humans (Gao et al., 2002). To increase transduction efficiency and change cellular tropism, vectors have been pseudotyped in which the ITRs of one serotype are packaged in the capsid of another serotype (Rabinowitz et al., 2002). For example, packaging AAV2 ITRs in an AAV8 capsid (generating a rAAV2/8 vector) confers the safety and efficient expression of rAAV2 with the tropism of AAV8. rAAV2/2 and rAAV2/8 are both able to transduce ganglion cells following intravitreal injection, and rAAV2/2, rAAV2/5 and rAAV2/8 are able to transduce photoreceptors (Lebherz et al., 2008).

Bipolar cells have been particularly difficult to target using AAV vectors, possibly as a result of viral particles being degraded within the retina before they access this layer. Modification of the capsid VP3 protein, specifically point mutation of tyrosine (Y) residues to phenylalanine (F), can significantly improve the ability of AAV to penetrate the retina and increase transgene expression (Zhong et al., 2008). A single mutation in rAAV2/8(Y733F) was demonstrated to result in bipolar cell transduction in the *rd10* (Doroudchi et al., 2011) and *rd1* mouse model of retinal degeneration (De Silva et al., 2016) and the rAAV2/2 quadruple and pentuple mutants are able to transduce bipolar cells when delivered either subretinally or intravitreally in wild-type mice (Petrs-Silva et al., 2011). A novel vector developed by targeted

mutagenesis of the AAV8 capsid, named AAV2/8BP2 has also shown efficacy in bipolar cell transduction (Cronin et al., 2014)

Conventionally, the AAV vector has been delivered via subretinal injection to target outer retinal cells (e.g. residual photoreceptors or bipolar cells) and via intravitreal injection to target ganglion cells. Subretinal injection is surgically more challenging. It is performed in the operating theatre and requires intraocular surgery to remove the vitreous (vitrectomy), as well as penetration of the retina using a fine gauge needle followed by sub-retinal injection of the virus. There are associated small risks of retinal detachment, bleeding, infection (end-ophthalmitis), inflammation, raised intraocular pressure and developing cataract; however, this method allows for concentrated vector delivery to an area directly adjacent to the residual outer retina, and is the method of injection for the Food and Drug Administration approved Luxturna (Maguire et al., 2019). Intravitreal injection is a commonly performed procedure in clinical care and can be performed in a suitable room in a clinical setting. It involves direct injection through the sclera at the pars plana, with subsequent injection of the drug into the vitreous cavity. However, dilution of virions within the vitreous cavity may reduce transduction.

More recently, a directed evolution approach has been utilised to generate new AAV variants, by screening a library of capsid types generated in the laboratory and isolating the best AAV candidate. Using this approach, an AAV vector named 7m8 was identified that enabled transduction of photoreceptors (Dalkara et al., 2013) and bipolar cells (Mace et al., 2015) following intravitreal delivery in mouse retina, with transduction of primate retina also being demonstrated in the former study. However, compared with intravitreal vector delivery in mouse models where widespread retinal transduction is often seen, a much smaller area of retina seems to be transduced in primates, largely in the parafoveal region (Ivanova et al., 2010; Yin et al., 2011). This reduction is presumed to be the result of a thicker inner limiting membrane in primates forming a greater barrier to virus particles than in the murine retina. Novel AAV vectors such as 7m8 do improve transduction in the primate retina (Byrne et al., 2020; Dalkara et al., 2013), although this is still not as widespread as in mouse models.

Alongside modification of the AAV capsid protein, alteration of the promoter can enable targeting of different cell populations. Many studies have used ubiquitous promoters to express the light sensitive protein in a wide range of retinal cells. An example of this is the CAG promoter [comprising a cytomegalovirus enhancer (C), chicken beta actin promoter (A) and splice acceptor of the rabbit beta globin gene (G)], which has been demonstrated to facilitate effective transduction of the degenerate retina (Bi et al., 2006;

Dalkara et al., 2013; De Silva et al., 2017). Subsequently, cell-specific promoters such as *Grm6-SV40* (the *grm6* gene being that which encodes the metabotropic glutamate receptor associated with ON bipolar cells) have successfully targeted expression to ON bipolar cells (Cehajic-Kapetanovic et al., 2015; Doroudchi et al., 2011; Lagali et al., 2008). The combination of this promoter with newer AAV capsid types further improves bipolar cell transduction (Mace et al., 2015) and modifications of the *Grm6-SV40* promoter have also been investigated, either using multiple repeats or additional intronic sequences (Cronin et al., 2014). Ganglion cell specific promoters such as the *synapsin-1* promoter (Berry et al., 2019; Sengupta et al., 2016) have been used to effectively target expression in these cell types.

## Clinical trials

More than a decade of research into optogenetic approaches for vision restoration has led to at least four ongoing clinical trials in this area (https://clinicaltrials.gov/ct2/results?cond=&term=optogenetic&cntry=&state=&city=&dist=). The patients most suitable for these therapies are those with advanced visual loss (visual acuity of hand movements or perception of light only) but with relatively intact inner retinal structures on optical coherence tomography (OCT) scans (Fig. 2). Primary outcome measures of these clinical trials include safety and tolerability, and secondary outcome measures include assessment of visual function such as perception of light, visual acuity, full field threshold stimulus test, determining direction of motion, identification of objects, evaluation of mobility and quality of life.

The first trial was led by RetroSense Therapeutics, now a part of Allergan, which began recruitment in 2015 (NCT02556736), investigating intravitreal delivery of ChR2 using an AAV2 vector in 14 patients. Few data have been released regarding outcomes: no serious adverse effects were reported, although one of 14 subjects was documented to have vitreal cells and one out of 14 subjects was documented to have vitritis; however, it is not clear whether these outcomes were noted in the same patient (https://clinicaltrials.gov/ct2/show/results/NCT02556736?term=channelrhodopsin&draw=2&rank=3). A phase 1/2a trial by GenSight Biologics named PIONEER (NCT03326336) commenced recruitment in 2017, and reported outcomes from the first patient in 2021 (Sahel et al., 2021). This study assessed intravitreal injection of an AAV2/7m8 vector, expressing a modified channel rhodopsin (ChrimsonR) under the control of a ubiquitous CAG promoter, in combination with a medical device incorporating light stimulating goggles. Patients were also given visual training using the goggles, beginning

4 months post-injection, to allow time for stabilisation of ChrimsonR in ganglion cells. The reported patient completed 15 such sessions and, 7 months after injection, showed an improvement in visual function such that they were able to perceive, locate and touch different objects using their treated eye alone whilst wearing goggles. Electroencephalographic recordings demonstrated activity in the visual cortex corresponding to these objects and visual perception. The patient could not detect any objects without goggles (or with the goggles prior to injection), indicating the need for light stimulus augmentation using this optogenetic tool. There was no evidence of intraocular inflammation in this patient following treatment.

Another clinical trial of optogenetic therapy uses an alternative channel rhodopsin named ChronosFP in combination with a prosthetic device (Bionic Sight LLC, NCT04278131). Further details regarding the vector have not yet been released, but pre-liminary reports via a press release indicate that four patients were able to detect light and motion starting 2–3 months following treatment with 'no safety concerns' reported (https://www.globenewswire.com/news-release/2021/03/30/2201412/0/en/First-Four-Patients-In-Bionic-Sight-s-Optogenetic-Gene-Therapy-Trial-Are-Able-To-Detect-Light-And-Motion.html). A fourth approach led by Nanoscope therapeutics investigates an intravitreal injection of an AAV2 based vector expressing a polychromatic opsin named multi-characteristic opsin (MCO) alone, that is without the use of a retinal prosthetic device. A phase 1/2 dose escalation study of 11 patients has been completed (NCT04919473) and a press release

on 1 year outcomes indicated an improvement in vision in all patients with regards to shape discrimination and mobility tests, with six out of seven patients in the high dose group reported to demonstrate gains in visual acuity (https://eyewire.news/articles/nanoscopes-optogenetic-gene-therapy-restores-clinically-meaningful-vision-in-11-patients-blinded-by-retinitis-pigmentosa/?c4src=article:infinite-scroll). No safety concerns were reported. A phase 2 trial (NCT04945772) has now recruited 27 patients and further details of these studies are awaited.

## Conclusions

Optogenetic therapies are one of several approaches in clinical trials for restoration of visual responses in advanced retinal degeneration. They have the advantage of being independent of the underlying disease-causing gene, and multiple reports have now demonstrated proof of principle in animal models (McClements et al., 2020; Simunovic et al., 2019). The first clinical trials are under-way in patients, with initial reports indicating safety, improved light sensitivity, and the ability to detect motion and larger objects.

However, several challenges remain, including the relatively low light sensitivity that is a feature of many ion channels or ion pumps used as optogenetic tools, resulting in the need for augmentation of the light stimulus via goggles or stimulating glasses. In addition, achieving high levels of transduction in the primate or human retina remains a challenge, and the ability to target specific retinal cell types, in an attempt to mimic normal retinal

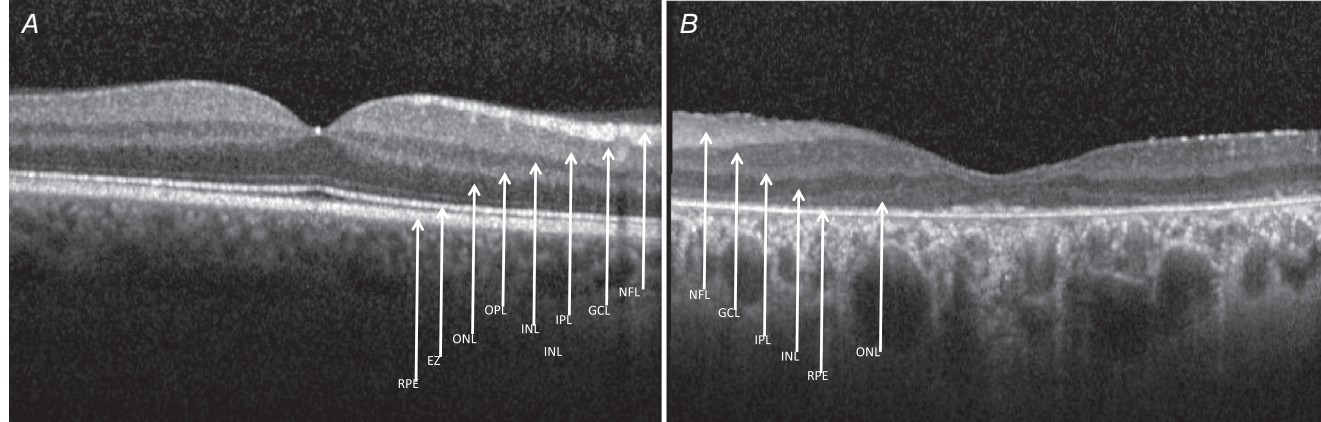

**Figure 2. Optical coherence tomography (OCT) in advanced inherited retinal degeneration**
OCT imaging measures reflectivity of low-coherence light to delineate retinal layers. Images shown are of a healthy retina (*A*) and from a patient with hand movements vision secondary to advanced retinal degeneration due to pathogenic variants in the *MERTK* gene (*B*). In the latter, inner retinal structures are largely intact. There is a small area of residual ONL (photoreceptor cell bodies) in (*B*), although the specialised inner and outer segments allowing phototransduction represented by the EZ have degenerated. Retinal layers: RPE, retinal pigment epithelium; EZ, ellipsoid zone (photoreceptor inner/outer segment junction); ONL, outer nuclear layer (photoreceptor cell bodies); OPL, outer plexiform layer; INL, inner nuclear layer; IPL, inner plexiform layer; GCL, ganglion cell layer; NFL, nerve fibre layer.

circuitry as closely as possible, is not yet well developed. How much precise targeting is required, and whether ubiquitous approaches are equally effective is currently unclear.

A further consideration is the role of retinal remodelling, which is a series of complex processes that occur in the degenerate retina. This includes glial hypertrophy, which may affect AAV penetration of the retina and transduction efficacy. Other changes include 'rewiring', in which neurons extend new processes that alter retinal circuitry, and 'reprogramming', where there is switching of bipolar cell classes (Pfeiffer et al., 2020). The impact of these altered cellular connections and circuitry on the potential visual response that can be restored via optogenetic methods is currently unknown, as well as whether cortical processing might adequately compensate for these changes.

Many of the answers to these questions may become clearer as we move further from bench to bedside since assessments of visual function and visual perception can be performed with far greater accuracy in patients. Knowledge generated in these trials may, in turn, lead to refinement of optogenetic tools and optimisation of image processing technology, with the aim of restoring vision in patients with advanced retinal degeneration.

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

# Additional information

## Competing interests

The authors declare that they have no competing interests.

## Author contributions

SDS and AM were responsible for the study conception and design. SDS was responsible for the literature search. SDS and AM were responsible for drafting manuscript. SDS and AM were responsible for revision of the manuscript. Both authors have read and approved the final version of this manuscript. All persons designated as authors qualify for authorship, and all those who qualify for authorship are listed.

## Funding

This work was funded by the Wellcome Trust (094448/Z/10/Z) and Fight for Sight (5099/5100).

## Keywords

gene therapy, inherited retinal degenerations, optogenetics

## Supporting information

Additional supporting information can be found online in the Supporting Information section at the end of the HTML view of the article. Supporting information files available:

**Peer Review History**

