## [Peer Review History · The Journal of Physiology]

Optogenetic approaches to therapy for inherited retinal degenerations

Samantha R Desilva and Anthony T Moore
DOI: 10.1113/JP282076

Corresponding author(s): Samantha Desilva (samantha.desilva@ndcn.ox.ac.uk)

The following individual(s) involved in review of this submission have agreed to reveal their identity: Neruban Kumaran (Referee #2)

Review Timeline:

Submission Date:	07-Mar-2022
Editorial Decision:	17-May-2022
Revision Received:	28-Jun-2022
Accepted:	18-Jul-2022

Senior Editor: Laura Bennet

Reviewing Editor: Omar Mahroo

Transaction Report:

Dear Dr Desilva,

Re: JP-SR-2022-282076 "Optogenetic approaches to therapy for inherited retinal degenerations" by Samantha R Desilva and Anthony T Moore

Thank you for submitting your invited Review-Symposium to The Journal of Physiology. It has been assessed by a Reviewing Editor and by 1 expert referee and I am pleased to tell you that it is considered to be acceptable for publication following satisfactory revision.

The reports are copied at the end of this email. Please address all of the points and incorporate all requested revisions, or explain in your Response to Referees why a change has not been made.

NEW POLICY: In order to improve the transparency of its peer review process The Journal of Physiology publishes online as supporting information the peer review history of all articles accepted for publication. Readers will have access to decision letters, including all Editors' comments and referee reports, for each version of the manuscript and any author responses to peer review comments. Referees can decide whether or not they wish to be named on the peer review history document.

I hope you will find the comments helpful and have no difficulty in revising your manuscript within 4 weeks.

Your revised manuscript should be submitted online using the links in Author Tasks Link Not Available. This link is to the Corresponding Author's own account, if this will cause any problems when submitting the revised version please contact us.

The image files from the previous version are retained on the system. Please ensure you replace or remove any files that have been revised. Your revised submission should include:

- A Word file of the complete text (including figure legends any Tables);
- An Abstract Figure (with legend in the Article file)
- Each figure as a separate, high quality, file;
- A full Response to Referees;
- A copy of the manuscript with the changes highlighted.
- Author profile. A short biography (no more than 100 words for one author or 150 words in total for two authors) and a portrait photograph of the two leading authors on the paper. These should be uploaded, clearly labelled, with the manuscript submission. Any standard image format for the photograph is acceptable, but the resolution should be at least 300 dpi and preferably more.

- A 'Cover Art' file for consideration as the Issue's cover image;
- Appropriate Supporting Information (Video, audio or data set https://jp.msubmit.net/cgi-bin/main.plex?form_type=display_requirements#supp).

To create your 'Response to Referees' copy all the reports, including any comments from the Reviewing Editor into a Word, or similar, file and respond to each point in colour or CAPITALS and upload this when you submit your revision.

I look forward to receiving your revised submission.

If you have any queries please reply to this email and staff will be happy to assist.

Yours sincerely,

Ian D. Forsythe
Deputy Editor-in-Chief
The Journal of Physiology
<https://jp.msubmit.net>
<http://jp.physoc.org>
The Physiological Society
Hodgkin Huxley House
30 Farringdon Lane
London, EC1R 3AW
UK
<http://www.physoc.org>
<http://journals.physoc.org>

REQUIRED ITEMS:

-Please include an Abstract Figure. The Abstract Figure is a piece of artwork designed to give readers an immediate understanding of the Review Article and should summarise the main conclusions. If possible, the image should be easily 'readable' from left to right or top to bottom. It should show the physiological relevance of the Review so readers can assess the importance and content of the article. Abstract Figures should not merely recapitulate other figures in the Review. Please try to keep the diagram as simple as possible and without superfluous information that may distract from the main conclusion of the Review. Abstract Figures must be provided by authors no later than the revised manuscript stage and should be uploaded as a separate file during online submission labelled as File Type 'Abstract Figure'. Please ensure that you include the figure legend in the main article file. All Abstract Figures will be sent to a professional illustrator for redrawing and you may be asked to approve the redrawn figure before your paper is accepted.

-Your MS must include a complete "Additional information section" with the following 4 headings and content:

Competing Interests: A statement regarding competing interests. If there are no competing interests, a statement to this effect must be included. All authors should disclose any conflict of interest in accordance with journal policy.

Author contributions: Each author should take responsibility for a particular section of the study and have contributed to writing the paper. Acquisition of funding, administrative support or the collection of data alone does not justify authorship; these contributions to the study should be listed in the Acknowledgements. Additional information such as 'X and Y have contributed equally to this work' may be added as a footnote on the title page.

It must be stated that all authors approved the final version of the manuscript and that all persons designated as authors qualify for authorship, and all those who qualify for authorship are listed.

Funding: Authors must indicate all sources of funding, including grant numbers. If authors have not received funding, this must be stated.

It is the responsibility of authors funded by RCUK to adhere to their policy regarding funding sources and underlying research material. The policy requires funding information to be included within the acknowledgement section of a paper. Guidance on how to acknowledge funding information is provided by the Research Information Network. The policy also requires all research papers, if applicable, to include a statement on how any underlying research materials, such as data, samples or models, can be accessed. However, the policy does not require that the data must be made open. If there are considered to be good or compelling reasons to protect access to the data, for example commercial confidentiality or legitimate sensitivities around data derived from potentially identifiable human participants, these should be included in the statement.

Acknowledgements: Acknowledgements should be the minimum consistent with courtesy. The wording of acknowledgements of scientific assistance or advice must have been seen and approved by the persons concerned. This section should not include details of funding.

EDITOR COMMENTS

Reviewing Editor:

This is a very clearly written symposium review discussing optogenetic therapies for inherited retinal dystrophies by experts in the field. The topic is introduced in context, and key issues relating to the development and deployment of these therapies, including the various alternative approaches taken, are insightfully discussed. The topic is of great importance as these diseases are currently largely untreatable and timely as breakthroughs in therapeutic approaches are now being made. Below are some suggestions for minor revisions to be considered by the authors (in addition to the reviewer comments).

Figures

The two figures are very helpful to the reader. An abstract figure is also needed (see guidance for symposium reviews) that

summarises the review (together with an abstract figure legend).

Also, perhaps an additional graphical element within the review would be helpful, either a table summarising current approaches or a figure (for example schematically depicting the AAV genome in the "Delivery of optogenetic agents to the retina" section) or both.

Abstract

"and are the leading cause of blindness in working age adults in England and Wales"

Since this journal has an international audience, consider changing to "and are a leading cause of blindness in working age adults in several countries."

"and benefit many more patients than gene-specific therapies."

Optogenetics is a type of gene therapy. It depends what you mean by gene-specific. Consider rephrasing as "and benefit many more patients than therapies directed at the specific gene implicated in each disorder."

Final sentence: "Alongside reviewing these different approaches, we discuss their application in four clinical trials that are now ongoing to investigate optogenetic therapies, with the aim of reversing visual loss in patients with end-stage retinitis pigmentosa."

To me, this sentence does not reflect balance of the article, as the 4 trials are discussed only in the final section. The authors might consider different phrasing, such as simply replacing this sentence with "Currently, at least four clinical trials are ongoing to investigate optogenetic therapies in patients, with the ultimate aim of reversing visual loss in end-stage disease."

Introduction

3rd paragraph: In contrast, optogenetics and retinal implants bypass host photoreceptors to stimulate remaining inner retinal neurons and postreceptoral pathways."

Could the authors add a sentence or two (and possibly a reference or two) regarding how the retinal implants work? Some readers will be unaware that they are electrical implants that electrically stimulate second or third order retinal neurons.

Ion channels

3rd paragraph: "... have peak sensitivity in the red light spectrum, which also may be beneficial given the increased tissue permeability of these wavelengths."

Since the optical media are transparent, is tissue permeability an issue? Or is it more that the longer (red) wavelengths undergo less scatter and could be less damaging than shorter (blue) wavelengths?

G-protein coupled receptors

2nd para: "in activation downstream signalling cascade..."

Should be "in activation of a downstream signalling cascade..."

"... and with responses" - consider deleting "and"

Target retinal cells

2nd paragraph: It is worth mentioning also that there are many types of amacrine cell and many types of ganglion cells.

Delivery of optogenetic agents to the retina

1st para: The tradename Luxturna is mentioned. Could mention the name of the active substance, voretigene neparvovec

4th para: "However, compared intravitreal vector..." Should probably read ""However, compared with intravitreal vector..."

REFeree COMMENTS

Referee #2:

This is a well written article on Optogenetics as a potential therapy for inherited retinal disease, that will be a useful addition to the current literature. Please find below suggested corrections:

In the Introduction section you state "gene-based therapies are likely to be effective early in the disease when significant numbers of viable photoreceptors are still present and will be ineffective when there is extensive retinal cell death." - Please expand why you think this to be the case? The clinical indication for Luxturna is the presence of photoreceptor structure on OCT regardless of extent. Perhaps consider changing ineffective to "less effective"?

In the last paragraph under the sub-heading Ion channels, should this be changed to read " Other native (e.g. ChrimsonR) and engineered (e.g. red-light activated depolarizing ChR or ReaChR) ChR have peak sensitivity in the red light spectrum, which also may be beneficial given the increased tissue permeability of these wavelengths.

In the last paragraph under the sub-heading Ion channels, please could you expand on studies conducted on the use LiGluR activated by a photoswitch - please expand on this in a few sentences.

Please expand on the red shifted cruxhalrhodopsin and archarrhodopsin studies.

Under the target retinal cells subheading you state: "In vivo studies have demonstrated successful targeting of eNpHR to hyperpolarise residual cones (Busskamp et al., 2012), and ChR2 to depolarise ON bipolar cells (Lagali et al., 2008; Doroudchi et al., 2011)." - Please expand on the nature of the in vivo study - eg mouse models.

Under the subheading Delivery of optogenetic agents to the retina you state "in human gene replacement therapy clinical trials and in an FDA approved retinal gene replacement therapy named Luxturna"- please consider changing this to gene supplementation rather than gene replacement.

Please also note Luxturna is both FDA and EMA approved.

Under the subheading Delivery of optogenetic agents to the retina you state that "Subretinal injection is surgically more challenging" - Please highlight the need for a vitrectomy with subretinal injection and the associated difficulties (including the need for an operating theatre) and associated risks and benefits.

Under the subheading Delivery of optogenetic agents to the retina you state that "Intravitreal injection is a commonly performed procedure" - Please clarify how this can be performed in a procedure room and associated risks and benefits.

Under the subheading of clinical trials you state the in clinical trial NCT02556736 demonstrated vitreous inflammation in 1 of 14 patients but the link suggests this is 2 out of 14 (1 patient vitreous cells and 1 vitritis)

Please also comment on safety of intravitreal injection of the AAV vector encoding ChrimsonR (i.e. no evidence of intraocular inflammation in the one patient.)

Please also comment on safety of the Bionic sight optogenetic therapy.

Please elaborate on trials for Nanoscope Therapeutics' MC0-10 optogenetic therapy- I understand data was presented from a completed phase 1/2a study in 11 patients and a new phase 2b trial (the trial you are referring) to has recruited 27 patients to date.

Please note in the Figure 2 legend there should be a Capital T in Tomography.

END OF COMMENTS

Confidential Review

07-Mar-2022

Response to editor and reviewer comments
JP-SR-2022-282076

EDITOR COMMENTS

Reviewing Editor:

This is a very clearly written symposium review discussing optogenetic therapies for inherited retinal dystrophies by experts in the field. The topic is introduced in context, and key issues relating to the development and deployment of these therapies, including the various alternative approaches taken, are insightfully discussed. The topic is of great importance as these diseases are currently largely untreatable and timely as breakthroughs in therapeutic approaches are now being made. Below are some suggestions for minor revisions to be considered by the authors (in addition to the reviewer comments).

We thank the editor for their comments and are grateful for the opportunity to revise this review.

Figures

The two figures are very helpful to the reader. An abstract figure is also needed (see guidance for symposium reviews) that summarises the review (together with an abstract figure legend). Also, perhaps an additional graphical element within the review would be helpful, either a table summarising current approaches or a figure (for example schematically depicting the AAV genome in the "Delivery of optogenetic agents to the retina" section) or both.

An abstract figure and legend have now been included. This also incorporates a schematic of the AAV genome and modes of delivery of the AAV vector to the retina, and therefore a further figure depicting this has not been incorporated.

Abstract

*"and are the leading cause of blindness in working age adults in England and Wales"
Since this journal has an international audience, consider changing to "and are a leading cause of blindness in working age adults in several countries."*

This change has been incorporated.

*"and benefit many more patients than gene-specific therapies."
Optogenetics is a type of gene therapy. It depends what you mean by gene-specific. Consider rephrasing as "and benefit many more patients than therapies directed at the specific gene implicated in each disorder."*

We agree with this comment and the change has been incorporated.

Final sentence: "Alongside reviewing these different approaches, we discuss their application in four clinical trials that are now ongoing to investigate optogenetic therapies, with the aim of reversing visual loss in patients with end-stage retinitis pigmentosa."

To me, this sentence does not reflect balance of the article, as the 4 trials are discussed only in the final section. The authors might consider different phrasing, such as simply replacing this sentence with "Currently, at least four clinical trials are ongoing to investigate optogenetic therapies in patients, with the ultimate aim of reversing visual loss in end-stage disease."

We agree with this comment and the change has been incorporated.

Introduction

3rd paragraph: In contrast, optogenetics and retinal implants bypass host photoreceptors to stimulate remaining inner retinal neurons and postreceptor pathways."

Could the authors add a sentence or two (and possibly a reference or two) regarding how the retinal implants work? Some readers will be unaware that they are electrical implants that electrically stimulate second or third order retinal neurons.

We agree and have added the following text: "The latter are arrays of electrodes or photodiodes that are surgically implanted in the subretinal space or on the surface of the retina permitting direct electrical stimulation of second or third order retinal neurons (Gekeler *et al.*, 2018; Bloch *et al.*, 2019)."

Ion channels

3rd paragraph: "... have peak sensitivity in the red light spectrum, which also may be beneficial given the increased tissue permeability of these wavelengths."

Since the optical media are transparent, is tissue permeability an issue? Or is it more that the longer (red) wavelengths undergo less scatter and could be less damaging than shorter (blue) wavelengths?

We agree with this comment and have changed the text to read "Other native (e.g. ChrimsonR) and engineered ChR (e.g. red-light activated depolarizing ChR or ReaChR) have peak sensitivity in the red light spectrum, and high light intensities of these wavelengths may be less damaging to the retina than similar intensities of blue light".

G-protein coupled receptors

2nd para: "in activation downstream signalling cascade..."

Should be "in activation of a downstream signalling cascade..."

"... and with responses" - consider deleting "and"

These changes have been incorporated.

Target retinal cells

2nd paragraph: It is worth mentioning also that there are many types of amacrine cell and many types of ganglion cells.

We agree and have inserted the following text in this paragraph: "There are at least 25 types of amacrine cell and over 17 types of ganglion cell in the primate retina, with varying functions that facilitate our complex visual responses (Grunert & Martin, 2020)."

Delivery of optogenetic agents to the retina

1st para: The tradename Luxturna is mentioned. Could mention the name of the active substance, voretigene neparvovec

This change has been incorporated

4th para: "However, compared intravitreal vector..." Should probably read ""However, compared with intravitreal vector..."

This change has been incorporated.

REFEREE COMMENTS

Referee #2:

This is a well written article on Optogenetics as a potential therapy for inherited retinal disease, that will be a useful addition to the current literature. Please find below suggested corrections:

We thank the reviewer for their kind comments.

In the Introduction section you state "gene-based therapies are likely to be effective early in the disease when significant numbers of viable photoreceptors are still present and will be ineffective when there is extensive retinal cell death." - Please expand why you think this to be the case? The clinical indication for Luxturna is the presence of photoreceptor structure on OCT regardless of extent. Perhaps consider changing ineffective to "less effective"?

We thank the reviewer for their comments and agree that Luxturna is indicated any stage when photoreceptor structure is preserved. The situation we refer to is in more advanced disease where extensive photoreceptor cell death or loss of photoreceptor structure is seen, and therefore the target for gene-specific replacement therapy is lost. We have clarified this in the text to read "Furthermore, gene-based therapies are likely to be effective early in the disease when significant numbers of viable photoreceptors are still present and are likely to be less effective in advanced disease when there is extensive photoreceptor cell death or photoreceptor structure has been lost."

In the last paragraph under the sub-heading Ion channels, should this be changed to read " Other native (e.g. ChrimsonR) and engineered (e.g. red-light activated depolarizing ChR or ReaChR) ChR have peak sensitivity in the red light spectrum, which also may be beneficial given the increased tissue permeability of these wavelengths.

This change has been incorporated.

In the last paragraph under the sub-heading Ion channels, please could you expand on studies conducted on the use LiGluR activated by a photoswitch - please expand on this in a few sentences.

Text has been inserted as follows: "Studies have also explored the use of an engineered light-gated ionotropic glutamate receptor (LiGluR). This bears a mutated cysteine residue, to which a photoswitchable tethered ligand can bind. Incident light of 380nm causes isomerisation of the photoswitch and opening of the ion channel, and 500nm light results in closing of the channel. Expression of this optogenetic tool in murine degenerate retina resulted in restoration of visual responses (Caporale *et al.*, 2011) and a second generation ligand activated at 460nm has also shown similar efficacy in mice and dogs (Gaub *et al.*, 2014)."

Please expand on the red shifted cruxhalrhodopsin and archarrhodopsin studies.

Further details on these studies have been included and this section now reads: "Other ion pumps have also been investigated such as the yellow-green light sensitive proton pump archaerhodopsin or Arch (Chow *et al.*, Nature 2010), activation of which leads to neuronal silencing. A further tool is Jaws, a red shifted cruxhalorhodopsin, which showed greater light sensitivity and ganglion cell spiking compared to other hyperpolarising ion pumps, following expression in residual cone photoreceptors in a mouse model of retinal degeneration (Chuong *et al.* 2014)."

Under the target retinal cells subheading you state: "In vivo studies have demonstrated successful targeting of eNpHR to hyperpolarise residual cones (Busskamp et al., 2012), and ChR2 to depolarise ON bipolar cells (Lagali et al., 2008; Doroudchi et al., 2011)." - Please expand on the nature of the in vivo study - eg mouse models.

The text has been clarified to state these studies were conducted using mouse models of retinal degeneration.

Under the subheading Delivery of optogenetic agents to the retina you state "in human gene replacement therapy clinical trials and in an FDA approved retinal gene replacement therapy named Luxturna"- please consider changing this to gene supplementation rather than gene replacement.

This change has been incorporated.

Please also note Luxturna is both FDA and EMA approved.

This has been incorporated in the text.

Under the subheading Delivery of optogenetic agents to the retina you state that "Subretinal injection is surgically more challenging" - Please highlight the need for a vitrectomy with subretinal injection and the associated difficulties (including the need for an operating theatre) and associated risks and benefits.

Under the subheading Delivery of optogenetic agents to the retina you state that "Intravitreal injection is a commonly performed procedure" - Please clarify how this can be performed in a procedure room and associated risks and benefits.

We thank the reviewer for their comments and have incorporated their suggestions. The text now reads: "Subretinal injection is surgically more challenging, given that it is performed in the operating theatre and requires intraocular surgery to remove the vitreous (vitrectomy), penetration of the retina using a fine gauge needle followed by subretinal injection of the virus. There are associated small risks of retinal detachment, bleeding, infection (endophthalmitis), inflammation, raised intra-ocular pressure and developing cataract, however, this method allows for concentrated vector delivery to an area directly adjacent to the residual outer retina, and is the method of injection for the FDA-approved Luxturna (Maguire et al., 2019). Intravitreal injection is a commonly performed procedure in clinical care and can be performed in a suitable room in a clinic setting. It involves direct injection through the sclera at the pars plana, with subsequent injection of the drug into the vitreous cavity and has fewer surgical risks, the main ones being infection (endophthalmitis), inflammation, raised intra-ocular pressure and a rare risk of cataract or retinal detachment. However, dilution of virions within the vitreous cavity may reduce transduction".

Under the subheading of clinical trials you state the in clinical trial NCT02556736 demonstrated vitreous inflammation in 1 of 14 patients but the link suggests this is 2 out of 14 (1 patient vitreous cells and 1 vitritis)

We thank the reviewer for this comment and agree that the data on clinicaltrials.gov indicates 1/14 patients with vitreous cells and 1/14 with vitritis. However it is not clear from the way the outcomes are reported as to whether these outcomes were noted in the same, or different patients (for example 11 ocular adverse effects are noted, but the total number of patients affected was 9). Therefore we have amended the text to read as follows: "...no serious adverse effects were reported but 1 of 14 subjects were documented to have vitreal cells and 1 out of 14 to have vitritis, however it is not clear if these outcomes were noted in the same patient

(<https://clinicaltrials.gov/ct2/show/results/NCT02556736?term=channelrhodopsin&draw=2&rank=3>)

Please also comment on safety of intravitreal injection of the AAV vector encoding ChrimsonR (i.e. no evidence of intraocular inflammation in the one patient.)

Text has been included to state: "There was no evidence of intraocular inflammation in this patient following treatment".

Please also comment on safety of the Bionic sight optogenetic therapy.

Formal data on safety have not yet been published but the press release indicated that there were "no safety concerns" in the 4 patients reported, and this information has been added to the manuscript.

Please elaborate on trials for Nanoscope Therapeutics' MCo-10 optogenetic therapy- I understand data was presented from a completed phase 1/2a study in 11 patients and a new phase 2b trial (the trial you are referring) to has recruited 27 patients to date.

We thank the reviewer for highlighting this omission and have now added text as follows:
" A phase 1/2 dose escalation study of 11 patients has been completed (NCT04919473), and a press release on 1 year outcomes indicated an improvement in vision in all patients with regards to shape discrimination and mobility tests, with 6 out of 7 patients in the high dose group reported to demonstrate gains in visual acuity (<https://eyewire.news/articles/nanoscopes-optogenetic-gene-therapy-restores-clinically-meaningful-vision-in-11-patients-blinded-by-retinitis-pigmentosa/?c4src=article:infinite-scroll>). No safety concerns were reported. A phase 2 trial (NCT04945772) has now recruited 27 patients and further details of these studies are awaited".

Please note in the Figure 2 legend there should be a Capital T in Tomography.

This change has been incorporated.

Dear Ms Desilva,

Re: JP-SR-2022-282076R1 "Optogenetic approaches to therapy for inherited retinal degenerations" by Samantha R Desilva
Anthony T Moore

I am pleased to tell you that your Symposium Review article has been accepted for publication in The Journal of Physiology, subject to any modifications to the text that may be required by the Journal Office to conform to House rules.

NEW POLICY: In order to improve the transparency of its peer review process The Journal of Physiology publishes online as supporting information the peer review history of all articles accepted for publication. Readers will have access to decision letters, including all Editors' comments and referee reports, for each version of the manuscript and any author responses to peer review comments. Referees can decide whether or not they wish to be named on the peer review history document.

The last Word version of the paper submitted will be used by the Production Editors to prepare your proof. When this is ready you will receive an email containing a link to Wiley's Online Proofing System. The proof should be checked and corrected as quickly as possible.

All queries at proof stage should be sent to tjp@wiley.com

The accepted version of the manuscript is the version that will be published online until the copy edited and typeset version is available. Authors should note that it is too late at this point to offer corrections prior to proofing. Major corrections at proof stage, such as changes to figures, will be referred to the Reviewing Editor for approval before they can be incorporated. Only minor changes, such as to style and consistency, should be made a proof stage. Changes that need to be made after proof stage will usually require a formal correction notice.

Are you on Twitter? Once your paper is online, why not share your achievement with your followers. Please tag The Journal (@jphysiol) in any tweets and we will share your accepted paper with our 22,000+ followers!

Yours sincerely,

Professor Laura Bennet
Senior Editor
The Journal of Physiology
<https://jp.msubmit.net>
<http://jp.physoc.org>
The Physiological Society
Hodgkin Huxley House
30 Farringdon Lane
London, EC1R 3AW
UK
<http://www.physoc.org>
<http://journals.physoc.org>

Comments:

Reviewing Editor:

The authors have largely addressed the editor/reviewer comments. This is a very readable review, summarising the field with useful insights.

Senior Editor:

Comments to the Author:

REFeree COMMENTS:

Referee #2:

Thank you for your point by point responses. I have nothing further to add.

*** IMPORTANT NOTICE ABOUT OPEN ACCESS ***

To assist authors whose funding agencies mandate public access to published research findings sooner than 12 months after publication The Journal of Physiology allows authors to pay an open access (OA) fee to have their papers made freely available immediately on publication.

You will receive an email from Wiley with details on how to register or log-in to Wiley Authors Services where you will be able to place an OnlineOpen order.

You can check if your funder or institution has a Wiley Open Access Account here <https://authorservices.wiley.com/author-resources/Journal-Authors/licensing-and-open-access/open-access/author-compliance-tool.html>

Your article will be made Open Access upon publication, or as soon as payment is received.

If you wish to put your paper on an OA website such as PMC or UKPMC or your institutional repository within 12 months of publication you must pay the open access fee, which covers the cost of publication.

OnlineOpen articles are deposited in PubMed Central (PMC) and PMC mirror sites. Authors of OnlineOpen articles are permitted to post the final, published PDF of their article on a website, institutional repository, or other free public server, immediately on publication.

Note to NIH-funded authors: The Journal of Physiology is published on PMC 12 months after publication, NIH-funded authors DO NOT NEED to pay to publish and DO NOT NEED to post their accepted papers on PMC.

1st Confidential Review

28-Jun-2022